

**Present-day high-resolution ice velocity map of the Antarctic ice sheet**
Qiang Shen[1,3], Hansheng Wang[1,3], C. K. Shum[2,1], Liming Jiang[1,3], Hou Tse Hsu[1,3], Jinglong
Dong[1,3], Song Mao[1,3], Fan Gao[1,3]
1. State Key Laboratory of Geodesy and Earth's Dynamics, Institute of Geodesy and
Geophysics, Chinese Academy of Sciences, Wuhan 430077, China
2. Division of Geodetic Science, School of Earth Sciences, The Ohio State University, Columbus,
Ohio 43210, USA
3. University of Chinese Academy of Sciences, Beijing 100049, China
*Correspondence to*: Qiang Shen (cl980606@whigg.ac.cn)
**Abstract:**
Ice velocity constitutes a key parameter for estimating ice-sheet discharge rates and is crucial
for improving coupled models of the Antarctic ice sheet to accurately predict its future fate
and contribution to sea-level change. Here, we present a new Antarctic ice velocity map at a
100-m grid spacing inferred from Landsat 8 imagery data collected from December 2013
through March 2016 and robustly processed using the feature tracking method. These maps
were assembled from over 73,000 displacement vector scenes inferred from over 32,800
optical images. Our maps cover nearly all the ice shelves, landfast ice, ice streams, and most
of the ice sheet. The maps have an estimated uncertainty of less than 10 m yr$^{-1}$ based on robust
internal and external validations. These datasets will allow for a comprehensive continent-
wide investigation of ice dynamics and mass balance combined with the existing and future
ice velocity measurements and provide researchers access to better information for



monitoring local changes in ice glaciers. Other uses of these datasets include control and
calibration of ice-sheet modelling, developments in our understanding of Antarctic ice-sheet
evolution, and improvements in the fidelity of projects investigating sea-level rise. All data
presented here can be downloaded from the Data Publisher for Earth & Environmental Science
(https://doi.pangaea.de/10.1594/PANGAEA.895738).



**1 Introduction**
Global warming could lead to significant mass changes in the Antarctic ice sheet. The ice mass
of this sheet has a displacement potential equivalent to a sea level rise greater than 60-m
(Fretwell et al., 2012;Alley et al., 2005), which would alter oceanic conditions and marine
ecosystems, such as ocean currents, water temperature, and fishing ground distributions (Gutt
et al., 2011). Monitoring the glacial dynamics of the ice sheet is a primary scientific goal to
determine whether the ice sheet is stable, growing or shrinking. Thorough and continued
monitoring of ice-sheet dynamics is also of utmost importance for accurate predictions of ice-
sheet behaviour in the future(Lucchitta and Ferguson, 1986). Ice velocity, which is one key
parameter representing ice dynamics, affects the estimates of ice-sheet mass balance and the
corresponding sea level rise(Scheuchl et al., 2012) and plays a crucial role in studies on glacier
dynamics and mass balance.  The ice velocity of peripheral outlet glaciers is one of the primary
parameters needed to determine the ice discharge rate, because these glaciers act as channels
for ice transportation from the ice-sheet interior to the ice shelves and ocean. A
comprehensive and lasting observation of ice velocity is important to better understand a wide
range of processes related to glacial mass fluxes, such as glacier response to climate and
climatic changes, glacier physics and flow modes, glacier flow instabilities (e.g., surges),
subglacial processes (e.g., erosion), and supra- and intra-glacial mass transport.



Ice velocity has been measured by traditional ground-based measurement techniques (e.g.,
GPS, electronic distance, aerial photograph) since the 1970s in the Antarctic ice sheet (Manson
et al., 2000; Zhang et al., 2008; Kiernan, 2001; Rott et al., 1998). However, obtaining a
complete real-time survey is difficult due to the remoteness of the continent and extremely
cold climate. Moreover, the sporadic and discontinuous measurements prohibit the study of
ice-sheet mass balance as a whole. Recently, glaciologists have begun to present a complete
picture of the ice velocity in Antarctica by using multi-satellite interferometric synthetic
aperture radar (InSAR) at a 450-m spatial resolution(Rignot et al., 2011). Additionally, an
updated dataset of annual InSAR-derived ice velocity was recently released at a 1000-m spatial
resolution, and another continent-wide ice velocity map from Landsat 8 (L8) images was also
reported(Mouginot et al., 2017) in a variety of spatial resolutions (300-1000 m). Long-term
and continuous measurements of ice velocity are a precondition for developing a complete
understanding of the ice dynamics of the continent of Antarctica. Furthermore, ice velocity
products with a higher resolution can facilitate more thorough investigations on localized ice
dynamics(Nath and Vaughan, 2003;Favier and Pattyn, 2015), such as the production of
crevasses and the role of ice rises on ice sheet stability. These factors highlight the need for a
new set of high-resolution ice velocity observations over Antarctica.
Deriving the surface velocity of glaciers and ice shelves using optical satellite images is a
rapid, cost-effective method to obtain the large-scale ice velocity field, especially in remote
Antarctica, which has been widely used in glaciology(Bindschadler and Scambos,
1991;Lucchitta and Ferguson, 1986;Burgess et al., 2013;Copland et al., 2009;Sam et al., 2016).
However, the Antarctic-wide ice velocity based on optical satellite images remains difficult to
determine, although relevant work has been performed since the mid-1980s. The near global
coverage and high repeat rate of optical satellites now provides the possibility for continent-
wide mapping and monitoring of glaciers and ice-sheet dynamics, especially the L8 mission. L8
is the newest generation of satellites in the Landsat family and provides continuous coverage



of earth's surface with a 16-day revisit cycle at a 98.2 inclination. The Operational Land Imager
(OLI) on L8 can provide improved radiometric and geodetic performance with a high spatial
resolution (up to 15 m). The combination of a high repeat rate and good performance creates
an opportunity to generate a continent-wide ice velocity map in Antarctica(Heid and Kääb,

81    2012).

82        Here, we present a high-resolution ice velocity mosaic of Antarctica (except for the area

south of 82.5°S) inferred from L8 images from the United States Geological Survey (USGS)
Earth Resources Observation and Science (EROS) Center. These velocity data have the highest
spatial resolution of 100 m achieved to date and were assembled from more than 73,000
scenes of displacement vectors. The vectors are inferred from more than 32,800 orthorectified
panchromatic band scenes with a 15-m spatial resolution acquired by the OLI on L8 from
December 2013 to March 2016 using the optical offset method (see Sect. 2). The flowchart for
producing and validating the ice velocity data is shown in Figure 1. These newly generated
datasets could be valuable in quantitative determination of ice discharge rates and mass
balance of the Antarctic ice sheet at present and contribute to climatic modelling studies.
Section 2 presents detailed information on extracting the ice velocity, including an ice velocity
generation method from displacement vectors and an error estimation approach. Section 3
presents the results and data records. Section 4 summarizes the accuracy validation process,
including the technical validation and internal validation using independent data. The last
section presents the conclusions.





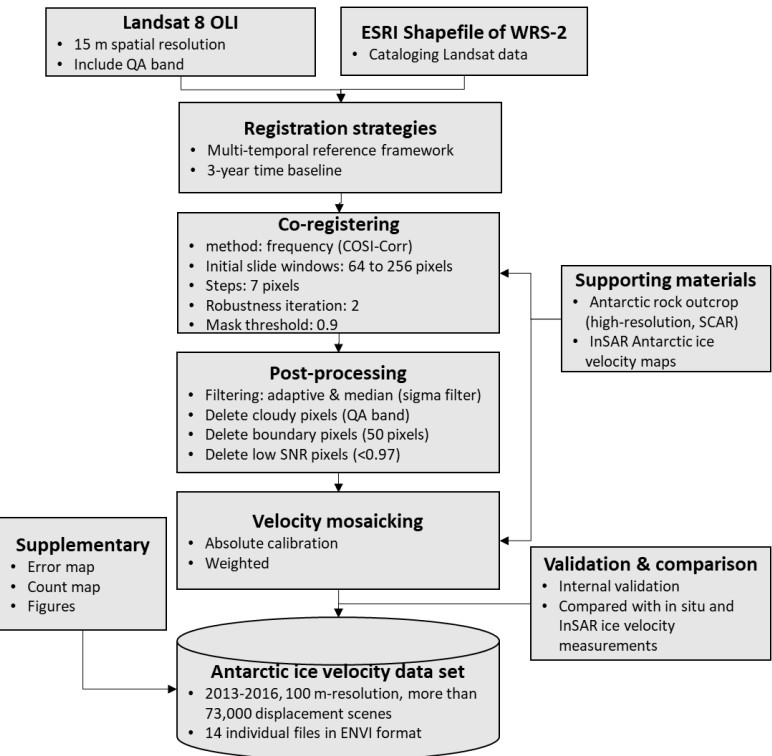


**Figure 1. Flowchart for producing and validating the generated L8 Antarctic ice velocity dataset.**

**2 Data and methods**

In this section, we first briefly present the satellite imagery data and existing ice velocity data collected using InSAR and field surveys (Sect. 2.1) and then summarize the pairing strategy of images (Sect. 2.2). Sections 2.3 to 2.5 summarize the feature tracking method of displacement, post-processing, ice velocity generation and mosaicking of Antarctic ice velocity maps.

**2.1 L8 imagery data and other independent ice velocity measurements**

In this study, the L8 orthorectified panchromatic band scenes are used to generate ice velocity maps using the optical offset method, which will be summarized in Section 2.3.(Leprince et al., 2007), and the quality assessment (QA) band provides a cloud ratio to identify the spatial distributions of clouds and water, which are masked in displacement scenes. In addition, the supporting data contain the InSAR-derived Antarctic ice velocity data, Antarctic rock outcrop



data inferred from L8, and previous ice velocity measurements compiled and managed by the
National Snow & Ice Data Center (NSIDC) and the Chinese Antarctic Center of Surveying and
Mapping. Antarctic rock outcrop data and the InSAR-derived Antarctic ice velocity data are
used to identify the stagnant region for absolute calibration and to assess our maps. The
existing measurements of ice velocity consist of satellite-derived measurements and in situ
measurements, which total over 144,000 measurements. The data include more than 1100 in
situ measurements from a variety of geodetic techniques, such as GPS, and electronic
distance, which provide an external validation of our ice velocity maps.
**2.2 Landsat 8 product and pairing strategy of images**
The L8 Level 1 Systematic Terrain Corrected (L1GT) products in GeoTIFF file format were
obtained from the USGS EROS Data Center (https://earthexplorer.usgs.gov/). The products
consist of ten 30-m spectral bands with coverage of visible, near infrared, and shortwave
infrared bands, a 15-m panchromatic band, product-specific metadata and a QA file. Here, we
only use the panchromatic band because of its high resolution, product-specific metadata and
QA file. More than 32,800 L1GT data have been processed to generate displacement vector
scenes. To overcome the cloud contamination and improve the amount of measurements, we
use the multiple reference strategy in image pairing, which means that all images in the same
Worldwide Reference System (WRS-2) could be taken as reference images in the pairings. In
addition, paired images are generated using a time interval of three years as a maximal
temporal baseline with a minimum time separation of 16 days. Finally, more than 73,000
paired images are obtained to produce the surface displacement of the ice sheet.
**2.3 Feature tracking processing**
To determine the horizontal displacement vectors due to ice motion, we use a feature tracking
method(Scambos et al., 1992;Bindschadler and Scambos, 1991;Leprince et al., 2007), also
known as the phase-shift method. The orthorectified L8 images are directly used to produce
the displacement vectors by means of the co-registration (or cross-correlation) method in the



Co-registration of Optically Sensed Images and Correlation (COSI-Corr) software package
developed at the California Institute of Technology(Leprince et al., 2007). Many studies have
proven that this technique is more efficient for images under different illumination
conditions(Heid and Kääb, 2012;Brown, 1992), especially in low visual contrast areas, such as
Antarctica. The method produces displacement vectors by a phase-shift technique of low
frequency calculated by a Fourier-based frequency correlator(Leprince et al., 2007), which is
produced repetitively within a specific sliding window (or patch) on the paired images. The
result is given by a three-band file consisting of an E-W displacement map (positive towards
the east), an N-S displacement map (positive towards the north), and a signal-to-noise ratio
(SNR) band as an indicator of the measurement quality. The technique can resolve sub-pixel
displacements of less than 1/20 of the pixel resolution at a high SNR, which is generally greater
than 0.9.
Specifically, the feature tracking processing has two stages. The first stage (namely, coarse
co-registration) approximately estimates the pixelwise displacement between two patches. In
general, if noisy images or large displacements are expected, a large initial sliding window
should be used. In this study, the size of the initial sliding window varies from 64 to 256 pixels
in both the X and Y directions according to a priori knowledge from the InSAR-derived Antarctic
ice velocity and the time interval between two paired images. Once the initial displacements
are estimated, the second stage consists of fine co-registration to retrieve the sub-pixel
displacement using a smaller window. The new size of 32×32 pixels is tentatively adopted to
yield reliable estimates for the displacement at densely independent points. Other parameters
of the frequency correlator include the step sizes between sliding windows in both the X and
Y directions (in pixels), the frequency masking threshold, the number of iterations for
robustness, resampling and gridded output. The step size is set to a constant value of 7 pixels
in each dimension or approximately a 100-m spatial resolution. A frequency masking threshold



of 0.9 is adopted as the optimum value as recommended in a previous study(Leprince et al.,

166   2007).

**2.4 Post-processing displacement vectors**

Generally, the frequency-based co-registration method is more accurate than statistical
methods but more sensitive to noise contamination. L8 images can minimize the effect
because of the good radiometric and geodetic performances, but decorrelation still exists due
to large ground motion, lack of measurable ground features (e.g., crevasses or rises), sensor
noise, illumination conditions, atmospheric changes (e.g., clouds) and topographic artefacts
(thereby producing imprecise orthorectified data). To overcome these problems, we devise
three steps to enhance the signal and exclude unreliable measurements. First, we suppress
the noise on each displacement scene using an adaptive filter and a median filter. The adaptive
filter is the local sigma filter(Eliason and McEwen, 1990) that features a filter size of 9 pixels
and a sigma factor value of 2. A median filter is further applied to remove "salt and pepper"
noise in ice displacement scenes. Second, the areas covered by clouds and water are excluded
from the displacement scenes using the QA band(Zanter, 2016). In the QA band, each pixel
contains a 16-bit integer that represents bit-packed combinations of surface, atmospheric, and
sensor conditions at different confidence levels. The pixels covered by cloud and water in
paired images are unpacked from the QA band using the procedures we have developed, and
the pixels marked as clouds and water at high confidence levels (67–100%) are used to build a
mask layer. These pixels are then masked in displacement scenes. Note that the identification
of cirrus clouds is problematic in raw images based on our analysis, since radiometric
characteristics of ice and cirrus clouds are generally indistinguishable. Here, we use only clouds
to build a mask layer. Third, since frequency correlation easily causes errors at the edges of
displacement scenes, the results of the displacement vectors are neglected at the edge
regions.

**2.5 Ice velocity measurement**



Cloud contamination is a major challenge in ice velocity estimation using optical images, which
is particularly significant in polar regions(Toon and Turco, 1991). To overcome this problem,
we process all image pairs using a time interval of three years as a temporal baseline with a
minimum repeat cycle of 16 days in WRS-2. Some images in adjacent paths in the WRS-2 are
also paired to determine the ice velocity for some void areas where no valid scenes with the
same path and row are available. A three-year time interval is used in our processing. Although
the decorrelation becomes more apparent with the increase in the time interval, many surface
features on ice sheets remain preserved and visible over many years(Lucchitta and Ferguson,
1986). Finally, more than 73,000 image pairs are organized from more than 32,800 scenes of
L8 panchromatic images and are processed to generate ice velocity estimates.
Despite the improved geometric accuracy of L8, the residual geolocation errors (~8 m
circular error (90%)) of the L8 panchromatic band contribute most of the uncertainties in ice
velocity products. These errors will lead to an offset between the displacement scenes and
should be removed(Fahnestock et al., 2016). In fact, offset tuning is often called absolute
calibration of the ice velocity data. In Antarctica, absolute calibration is a challenging issue
because the ice is active almost everywhere and available rock outcrops are extremely scarce.
Here, we use both rock outcrop data (Figure 3e) derived from L8 images(Burton-Johnson et
al., 2016) and the InSAR-derived Antarctic velocity map(Mouginot et al., 2017) to determine
the relatively stagnant areas (i.e., areas with InSAR-derived ice velocities of <10 m yr$^{-1}$) for the
absolute calibration of our ice velocity estimates.
The velocity calibration consists of three steps. First, the differences in the displacements
between the InSAR-derived velocity map and our calculated ice velocity maps from Landsat
images are calculated in the stagnant areas. Second, to eliminate outliers, a $3\sigma$ filter is
applied recursively on these differences. In this technique, the measurements of the
differences are removed if the magnitudes of the values are larger than three times the
standard deviation ($3\sigma$). Third, the mean of the remaining differences is considered the offset



of the displacement scenes. Furthermore, the offsets for the displacement scenes outside of
the stagnant areas (such as in the Ross and Ronne ice shelves) are estimated by overlapping
neighbouring scenes captured at approximately the same time. The offsets of two velocity
components are independently estimated. In addition, to be computationally efficient,
Antarctica is divided into 16 gridded sub-regions, which are shown in Figure 2a, and data
stacking is processed independently. Finally, the 14 sub-regions are mosaicked to generate an
ice velocity map for all of Antarctica because two sub-regions do not cover the grounded ice
sheet.

225        The mosaicked velocity maps are produced based on the displacement scenes. To increase

the accuracy of the mosaicked velocity maps, we stack all displacement scenes after removing
the pixels with an SNR less than 0.95. In general, the velocity map contains dozens of scenes
in each location. For a specific pixel denoted by $i$, all displacement scenes ($m$=1, 2, …, $n$) are
stacked to obtain the estimate of the ice velocity ($V_i$) as follows:
$$V_i = \frac{\sum_{m=1}^{n} \Delta d_m^i}{\sum_{m=1}^{n} \Delta t_m^i} \qquad\qquad (1)$$

where $\Delta d_m^i$ denotes the generated displacement during the time interval $\Delta t_m^i$.
**3 Results and validations**
**3.1 Antarctic-wide ice velocity map**
Over Antarctica, valuable L8 images are available for only the summer and fall seasons, i.e., in
November, December, January, February and March, which means that the L8 ice velocities
represent mainly the summer/fall ice velocity. In Figure 2a, we show a mosaicked Antarctic ice
velocity map inferred from over 73,000 L8 images acquired from December 2013 to March
2016. Our maps cover nearly all of the ice shelves, ice streams, and the majority of the ice
sheet. Here, Figure 2b shows a count map to indicate the number of images used to produce



the ice velocity data. The predominant year of the images is 2015 (Figure 2b), and there are
generally more than 20 displacement vectors, with up to 200 in some regions. The L8 ice
velocity map shows the same pattern as the InSAR-derived ice velocity map, and Figure 3
shows some ice velocity and the difference graphs from the two ice velocity products. The
spatial resolution of the L8 ice velocity data is 3 to 10 times finer or higher resolution than that
of the recent L8-derived(Gardner et al., 2018) and InSAR-derived ice flow maps(Mouginot et
al., 2017;Rignot et al., 2011), reaching up to 100 m. Here, we show the velocity map of James
Ross Island in the Antarctic Peninsula as an illustration of our high-resolution results (Figures
2c and 2d). The results reveal that the L8 velocity map can provide details of the ice velocity
pattern for the Antarctic ice sheet, such as for James Ross Island and small glaciers (Figures 2c
and 2d). Thus, our ice velocity map provides the first opportunity to investigate localized ice
dynamics, such as crevasse formation, and the roles of ice rises and rumples in ice-sheet
dynamics and evolution. These maps also have good coverage over Antarctica, except for
south of 82.5°S. The mosaicked ice velocity map covers the majority of the Antarctic ice sheet
and nearly 99% of the fast-flowing glaciers and ice shelves, as well as fast ice, except for a few
ice streams located on the Ronne Ice Shelf (e.g., Academy and Foundation Glaciers) and on
the Ross Ice Shelf (e.g., Whillans Glacier in the Siple Coast).

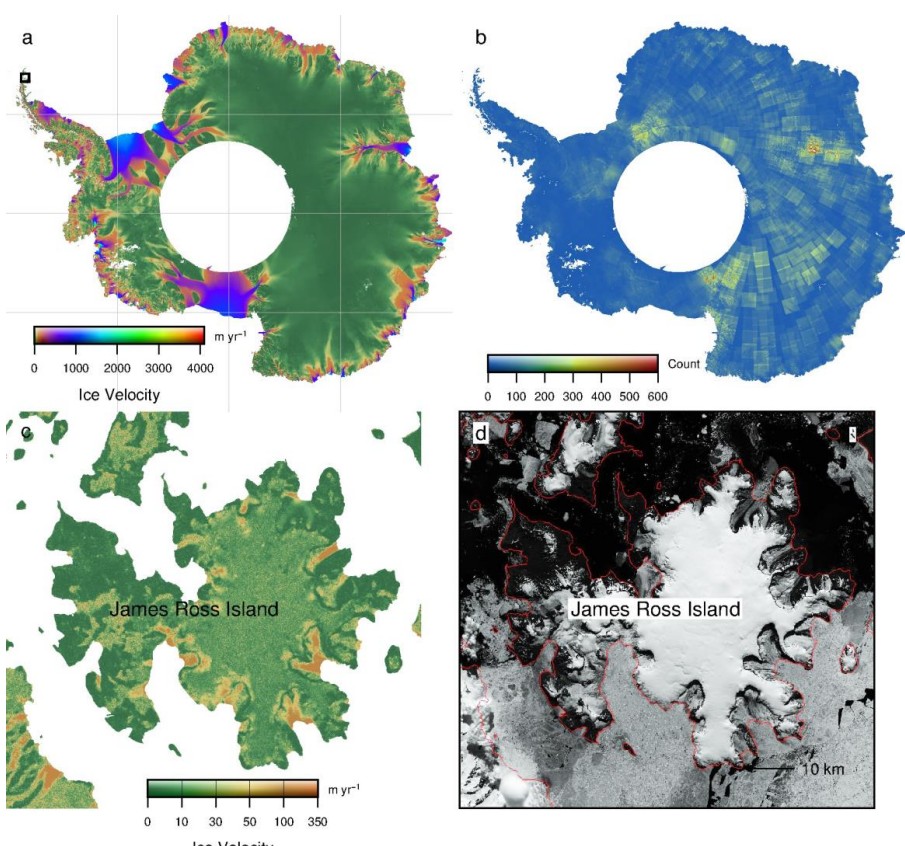


**Figure 2. L8-derived ice velocity estimates from Landsat 8 images from December 2013 to**

**March 2016. (a)** L8-derived Antarctic ice velocity map (gridded lines delineate the 16 sub-

regions); **(b)** footprint map of L8 presenting the number of valid displacement vectors used to

produce the ice velocity map in a specific grid (pixel); **(c)** magnified view of the ice velocity map

of James Ross Island, Antarctic Peninsula, corresponding to the box in Figure 2a; and **(d)** L8

image corresponding to Figure 2c, in which the red solid line shows the coastal lines. The L8-

derived ice velocity maps are drawn on a 500-m grid. The maps were created using The Generic

Mapping Tools (http://gmt.soest.hawaii.edu/), Version 5.2.1.(Wessel et al., 2013)

267

268

269





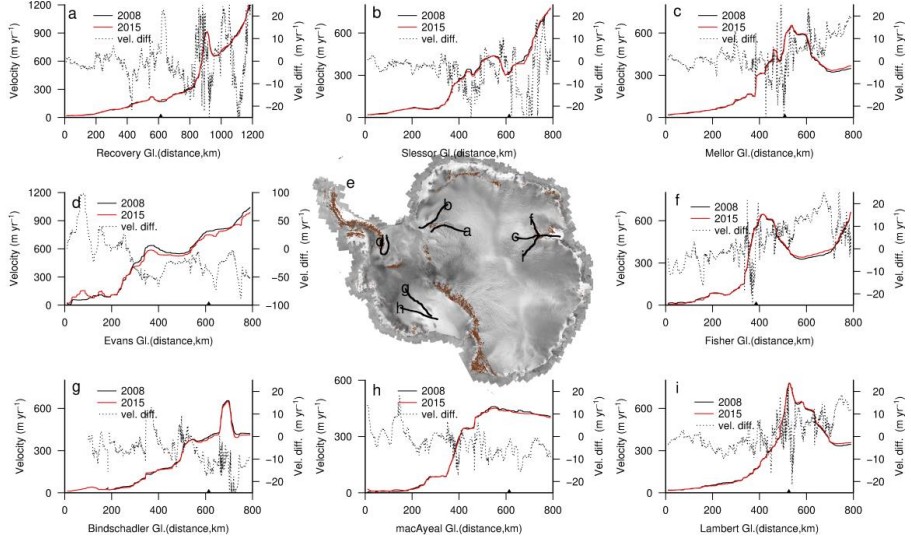

**Figure 3. Velocity profile and the difference graphs comparing the L8 (2015) and InSAR (2008)**

**ice velocity data.** (**a**) Recovery Glacier; (**b**) Slessor Glacier; (**c**) Mellor Glacier; (**d**) Evans Glacier;

(**e**) AMM RAMP Antarctic mosaic(Jezek and Team, 2002), in which the brown area shows the

areas covered by rock outcrop. The black solid lines and the letters show the geolocations of

the velocity profiles in a-i; (**f**) Fisher Glacier; (**g**) Bindschadler Glacier; (**h**) MacAyeal Glacier; and

(**i**) Lambert Glacier. Left y-labels represent velocity magnitudes of glaciers, and the right y-

labels are the differences of velocity magnitudes along sampling paths shown in figure **e**. Black

triangles represent the location of grounding lines. The maps were created using The Generic

Mapping Tools (http://gmt.soest.hawaii.edu/), Version 5.2.1.(Wessel et al., 2013)

**3.2 Data Records**

The     ice     velocity     map     and     supporting     data     are     archived     at

https://doi.pangaea.de/10.1594/PANGAEA.895738. The file format used is the ENVI Standard.

Examples of the data products are shown in Figure 2.

**3.2.1 Ice velocity map and error map**



The 100×100-m gridded ice velocities for all of Antarctica are stored in a 16-bit long point ENVI
file format under a polar stereographic projection with a true latitude of 71°S. The gridded ice
velocity has been equally divided into 4 subsets in the X and Y directions in consideration of
file size and computer processing speed (Table 1). Note that ENVI file sizes have been kept to
approximately 1 Gigabyte for user friendliness and easy downloading. Each ENVI file contains
three bands that show velocity vectors in both the X-direction and Y-direction and a gridded
error map of the ice velocity. The structure of the ice velocity filenames is
Velocity_l8_*year*_*subset*_*XY*.dat, where Velocity represents ice velocity data and l8
indicates the L8 satellite from which images are used to produce the ice velocity map. *year*
is the predominant year of the images in the file, *subset* shows whether the ice velocity file
has been cropped due to considerations of file size and computer processing speed, and *XY*
indicates the relative coordinates among all files, where X is the column number starting with
one and Y is the row number starting with one. Each ENVI file is associated with a head file
with the same filename. The head file contains the coordinate information (see xstart and
ystart) for the subset among the Antarctic gridded ice velocity data.
**Table 1. Filename structure of the ice velocity ENVI files. Two tiles do not include any valid**
**ice velocity values.**

| Ice velocity filenames | column | row | year | subset |
|---|---|---|---|---|
| Velocity_l8_2015_1_11.dat | 1 | 1 | 2015 | 1 |
| Velocity_l8_2015_2_12.dat | 2 | 1 | 2015 | 2 |
| Velocity_l8_2015_3_13.dat | 3 | 1 | 2015 | 3 |
| Velocity_l8_2015_4_14.dat | 4 | 1 | 2015 | 4 |
| Velocity_l8_2015_5_21.dat | 1 | 2 | 2015 | 5 |
| Velocity_l8_2015_6_22.dat | 2 | 2 | 2015 | 6 |
| Velocity_l8_2015_7_23.dat | 3 | 2 | 2015 | 7 |
| Velocity_l8_2015_8_24.dat | 4 | 2 | 2015 | 8 |



| Velocity_l8_2015_9_31.dat | 1 | 3 | 2015 | 9 |
|---|---|---|---|---|
| Velocity_l8_2015_10_32.dat | 2 | 3 | 2015 | 10 |
| Velocity_l8_2015_11_33.dat | 3 | 3 | 2015 | 11 |
| Velocity_l8_2015_12_34.dat | 4 | 3 | 2015 | 12 |
| Velocity_l8_2015_13_41.dat | Not provided. Subset does not cover the grounded ice sheet. | | | |
| Velocity_l8_2015_14_42.dat | | | | |
| Velocity_l8_2015_15_43.dat | 3 | 4 | 2015 | 15 |
| Velocity_l8_2015_16_44.dat | 4 | 4 | 2015 | 16 |


**3.2.2 Landsat ground footprints**

Landsat gridded footprints are stored in 8-bit integer point ENVI files (Figure 2b), which show
the number of displacement vectors used to produce the ice velocity at a specific location.
These files also have the same file structure and projection as the gridded ice velocity map.
The naming convention of the footprint maps is Footprints_l8_*year*_*subset*_*XY*.dat,
which has the same naming convention as the ice velocity maps, except for "Footprints", which
indicates the content of the product.

**4 Technical Validation**

Verification of the continent-wide ice velocity in the Antarctic ice sheet is the most difficult
task in the absence of other independent measurements, which are difficult to obtain because
of the remoteness of the continent and the harsh climate in Antarctica. Here, we describe and
assess the precision by internal validation and comparison with in situ measurements. For
internal validation, we produce the gridded error maps for the velocity maps. Furthermore,
we compared our velocity maps with the InSAR-derived ice velocity map and in situ



measurements as well as pre-existing measurements from remote images using co-
registration vectors.

**4.1 Internal validation**
In the absence of other synchronously independent measurements of ice velocity, the
uncertainty in the ice velocity maps from empirical analysis is generally used as an estimate of
the accuracy of the ice velocity product. The error sources of L8-derived ice velocity are
primarily attributed to the following three aspects: image co-registration, paired image time
interval, and stacked data quantity.
Image co-registration represents a process of geometrically aligning two or more satellite
images to obtain the corresponding pixels or feature representing the same surface objects,
which is a main factor that influences the ice velocity accuracy. The image co-registration
accuracy is largely determined by the following three factors: (1) decorrelation due to dramatic
ground changes and a lack of measurable features between the scenes due to long time
intervals or low-contrast land cover (e.g., snow or ice); (2) low image quality caused by sensor
noise, pixel oversaturation, aliasing and cloud contamination; and (3) topographic artefacts
caused by shadowing differences and inaccurate orthorectification of satellite attitudes. In
fact, quantifying the effects of the three error sources is very difficult, especially on a pixel-by-
pixel scale. In general, the co-registration accuracy is given empirically based on the validation
of the matching algorithm. Here, the co-registration accuracy is equal to 1/10 of the pixel size
in the E-W and N-S displacement components. This value is greater than 1/50 of the pixel size
proposed by Leprince *et al*. (2007)(Leprince et al., 2007).
The second factor is the time interval of the paired images, because ice velocity is a function
of displacements and time. Ice velocity is calculated by the displacement divided by the time
interval of paired images. The uncertainty of the displacement is primarily attributed to image





co-registration as mentioned above. Thus, a longer time interval suggests higher precision of
the ice velocity (see Eq. 2).
The third factor is the amount of stacking data. Hence, more displacement data are stacked
and will be more accurate. As a result, the gridded error map of ice velocity data can be
obtained pixel by pixel based on the method of error propagations using the co-registration
accuracy, time interval and total amount of stacking data.
According to the mosaicking method discussed above (Eq. 1), the uncertainty in one
mosaicked velocity component at the $i$-th pixel (denoted by $\sigma_{V_i}$) can be estimated using the
following error propagation formula under the assumption that the errors from different
sources are independent and any temporal errors are negligible:
$$\sigma_{V_i} = \pm \sqrt{ \sum_{m=1}^{n} (\sigma_m^i)^2 \Big/ \left( \sum_{m=1}^{n} \Delta t_m^i \right)^2 } \qquad (2)$$
where $\sigma_m^i$ is the co-registration error, i.e., the standard deviation of the $m$-th displacement
observation during the time interval $\Delta t_m^i$. Since the co-registration errors are constant in the
spatial (the whole scene) and temporal domains (all stacked displacements), if $\sigma_m^i$ is assumed
to be a constant of $\sigma$, Equation (2) can be simplified as follows:
$$\sigma_{V_i} = \pm \sqrt{n}\,\sigma \Big/ \sum_{m=1}^{n} \Delta t_m^i \qquad (3)$$
Since the E-W and N-S components at the $i$-th pixel have the same uncertainty, which can be
calculated with Equation (3), the uncertainty is valid for the magnitude of the velocity vector.
The error map (Figure 4a) in the magnitude of the mosaicked velocity vector is generally better
than 10 m yr$^{-1}$, except for some areas in the Antarctic Peninsula and Marie Byrd Land in West
Antarctica. Fewer valid satellite images are obtained from the two regions due to heavy cloud
coverage. Relatively large uncertainties in these areas were mainly caused by a small amount
of valid displacement vectors. For comparison, the error map of the InSAR ice velocity



estimates is shown in Figure 4b, which quantifies the achieved accuracy of the ice velocity
maps of this study.

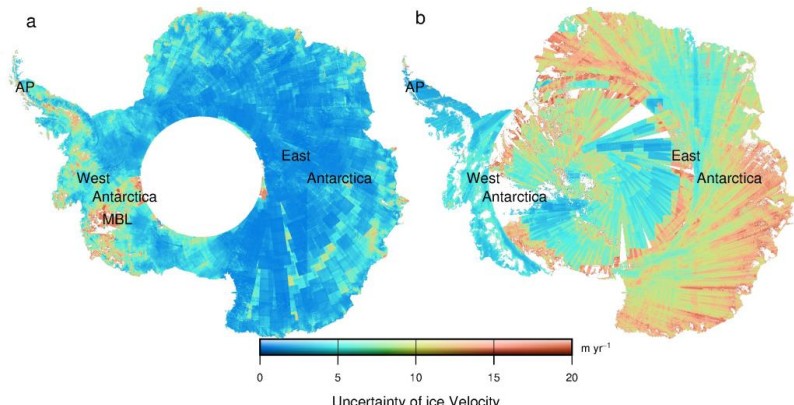


**Figure 4.** Uncertainty maps of the L8-derived Antarctic ice velocity (**a**) and InSAR-derived ice
velocity(Mouginot et al., 2017) (**b**) MBL: Marie Byrd Land. The maps were created using The
Generic Mapping Tools (GMT, http://gmt.soest.hawaii.edu/), Version 5.2.1.(Wessel et al.,

377  2013)


**4.2 Comparisons with other datasets and in situ measurements**

A comparison of our velocity measurements with previous velocity measurements would
be very beneficial. However, this comparison is very difficult due to the variability of glacier
flow. Some glacier flow may vary significantly on daily, seasonal and yearly scales. Here, we
collected historical long-term ice velocity measurements compiled and managed by the
project of Velmap(Raup and Scambos). Our ice velocity results are compared with only the in
situ measurements located in the slow-flowing areas (<100 m yr$^{-1}$) because highly dynamic
changes in ice velocity in fast-flowing areas (e.g., ice shelf) are expected. Furthermore, the ice
velocity measurements on Byrd Glacier determined by photogrammetric methods and on
Amery ice shelf from theodolite/EDM and GPS methods are used to illustrate the performance
of our ice velocity map(Brecher, 1982;Brecher, 1986;Allison, 1979). A total of 609 sites in slow-



flowing areas were chosen for comparison and analysis, and their differences are shown with
dots in Figure 5a, where the colours of the dots denote the magnitude of the differences.
Figure 5b shows the histogram of the differences between our velocity data and the in situ
measurements. Except for three sites (two in Lambert-Amery Basin in East Antarctica and one
on the Siple Coast in West Antarctica) (Figure 5a), the points are all less than 40 m yr$^{-1}$, and
593 sites, representing more than 97% of the total check points, have differences in the ice
velocity magnitude of less than 10 m yr$^{-1}$. The differences have a $-0.7$ m yr$^{-1}$ mean value and a
3.2 m yr$^{-1}$ standard deviation. For comparison, the differences between the InSAR velocity and
field surveying data are shown in Figure 5c. A total of 589 points are less than 10 m yr$^{-1}$. These
points have a mean value of 0.3 m yr$^{-1}$ and a standard deviation of 3.3 m yr$^{-1}$. To further
investigate the performance of the L8 ice velocity data in slow-flowing areas, we compared
the L8, InSAR and in situ measurements with ice velocity magnitudes of less than 20 m yr$^{-1}$.
The analysis results are shown in Figure 6. Figures 6a and 6b show the difference between the
L8 ice velocity data and in situ measurements and the statistical results of these differences.
The near-zero y-intercept and nearly unitary slope of the data in Figure 6a confirm that the L8
ice velocity data are very consistent with the in situ measurements, even in the stable interior
ice sheet. Figure 6c shows the InSAR results and the similar performance with the L8 ice
velocity data. In summary, the L8 ice velocity data have an accuracy of 10 m yr$^{-1}$ ($3\sigma$).



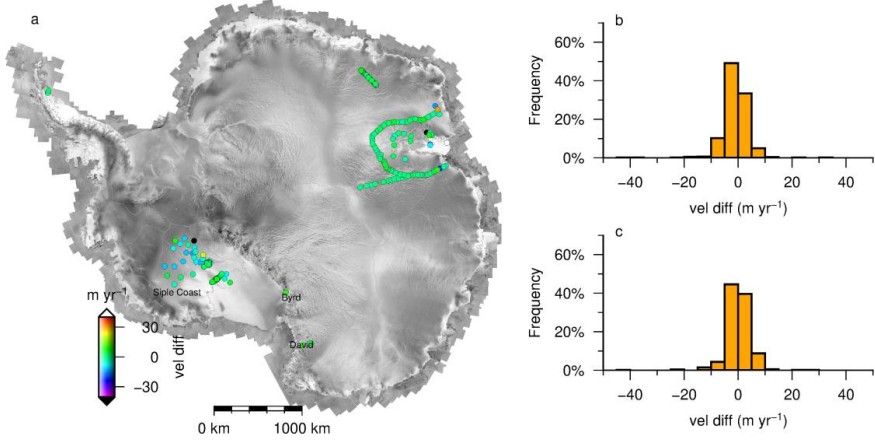



**Figure 5. Accuracy assessment of L8 ice velocity and InSAR velocity. (a)** Differences between

in situ measurements and L8 ice velocity, where the colour dots show the geolocations and

velocity differences between the L8 and in situ measurements (background maps are from the

AMM RAMP Antarctic mosaic(Jezek and Team, 2002)); **(b)** histogram of the differences

between the L8 and in situ ice velocity data; and **(c)** histogram of the differences between the

InSAR and in situ ice velocity data.


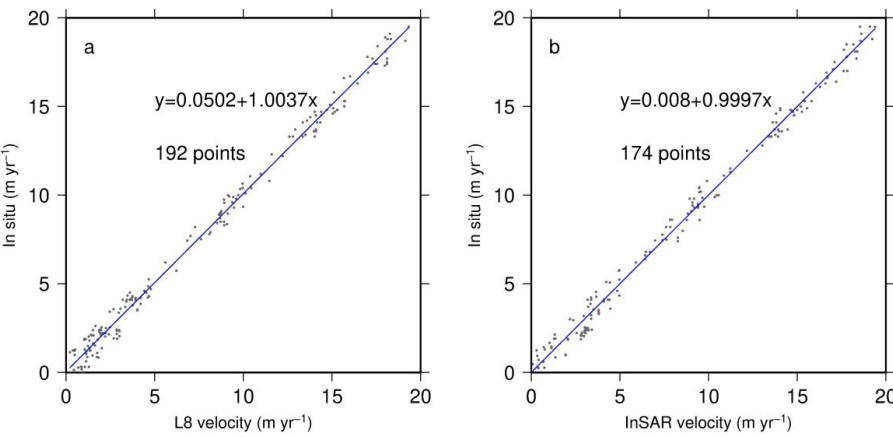


**Figure 6. Comparison between the in situ measurements and L8 ice velocity (a) and InSAR**

**(b)**






To assess the reliability of the L8 ice velocity data in fast-flowing areas, here, we show two
illustrations of Byrd Glacier (80°S, 160°E) and Amery ice shelf (69°S, 71°E). For Byrd Glacier,
the in situ ice velocity measurements were determined by photogrammetric methods from
two sets of aerial photographs acquired on 6 December 1978 and 21 January 1979(Brecher,
1982). In total, ice velocities of 470 sites on the main ice stream were determined by the
change in location of natural features over the 56-day interval between two flights. Here, we
analysed 436 sites where the velocity is greater than 100 m yr$^{-1}$. The near unitary slope of
Figure 7b shows that L8 and the aerial ice velocity data have a good correlation, except for a
small number of sites that have a relatively large difference. Figure 7a shows that the sites
with large differences are located mainly on the lateral side of the glacier (see red and dark
blue dots in Figures 7a and 7b). The large difference may be caused by the following factors:
(1) errors in the two datasets, (2) relatively low resolution of the L8 ice velocity relative to the
aerial method and high-velocity gradient on the sides of the glacier, which easily causes large
differences, and (3) velocity changes between the long time intervals. For comparison, we also
show the InSAR and aerial ice velocity in Figure 7c, which shows that the L8 and InSAR ice
velocity in fast-flowing areas have the same performance as those shown in Figures 7b and 7c.
On Amery ice shelf, the in situ ice velocity measurements were determined by using a
combination of standard surveying techniques, including electronic distance and theodolites
and GPS. The ice velocity observations on Amery ice shelf were mainly collected during two
time periods (December 1968—January 1970; December 1988—January 1991). Finally, the ice
velocity measurements of 120 sites were compared with the L8 and InSAR ice velocity data.
The vast majority of differences are less than 200 m yr$^{-1}$, except for a small number sites
beyond the range (Figure 7d). The sites with large differences are mainly located on the front
of the ice shelf. The L8 ice velocity data agree well with the in situ measurements in Figure 7e.
For comparison, the InSAR and in situ measurements are also shown in Figure 7f. An increase
in the ice velocity is observed between the two time periods along the lateral route at an
average velocity of 800 m yr$^{-1}$ (Figures 7d and 7e), and this phenomenon is also shown in Figure
7f. The apparent changes in ice velocities may suggest different patterns in the dynamic
characteristics.

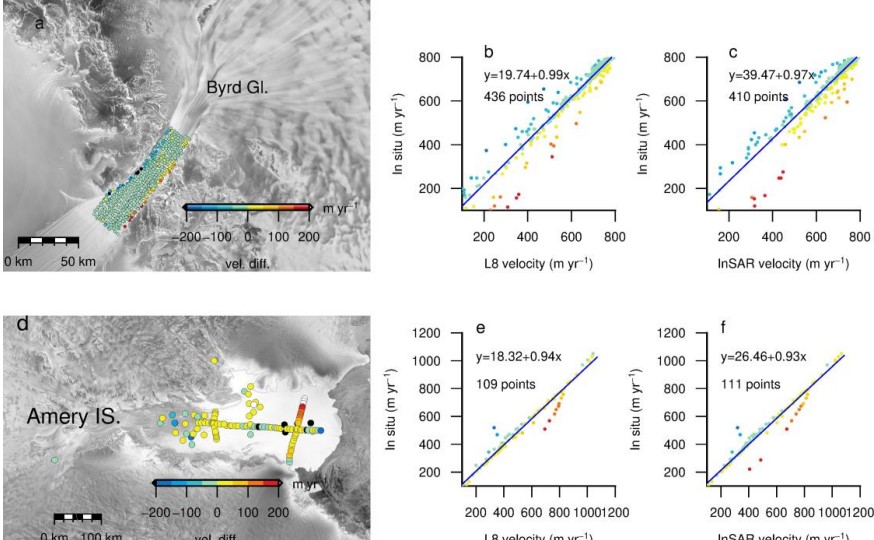

**Figure 7. Comparison among the L8, InSAR and in situ measurements.** (a) Byrd Glacier and
the differences between the L8 and in situ ice velocity data (colour dots); note that the black
dots represent differences of less than -200 m yr$^{-1}$, and the white dots represent differences
of greater than 200 m yr$^{-1}$, (b) comparison between the L8 and in situ measurements on Byrd
Glacier, (c) comparison between the InSAR and in situ measurements on Byrd Glacier; (d)
Amery ice shelf and the differences between the L8 and in situ ice velocity data (colour dots),
(e) comparison between the L8 and in situ measurements on Amery ice shelf, and (f)
comparison between the InSAR and in situ measurements on Amery ice shelf.
**5 Conclusions**
Cold regions are very sensitive to the impacts of climate change. Long-term monitoring of ice-
sheet dynamics is crucial for precise assessments of the glacial responses to climate change.
We constructed a new Antarctic-wide high spatial resolution ice velocity map inferred from

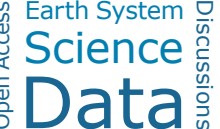

Landsat 8 imagery data collected between 2013 and 2016. The new map will provide a
opportunities to comprehensively investigate Antarctic ice dynamics in combination with
existing and future ice velocity maps, which will provide insights into the ice sheet's mass
balance.
**Data availability**
The latest dataset is available at the Data Publisher for Earth & Environmental Science
(https://doi.pangaea.de/10.1594/PANGAEA.895738)

**Author contributions**
Q.S. conceived of, designed and conducted the experiment. H.W. contributed to the
research framework and helped develop the methodology. C.K.S. and L.J. performed
the data analysis. H.T.H. contributed to analysing the results. J.D., S.M. and F.G.
contributed to the data processing. All authors contributed to the discussion and writing
of the manuscript.
**Competing interests**

The authors declare that they have no competing interests.


**Acknowledgements**
We thank the National Aeronautics and Space Administration (NASA) and United
States Geological Survey (USGS) for providing the Landsat-8 data. We thank E. Rignot
and Alex S. Gardner at the Jet Propulsion Laboratory/California Institute of Technology
for providing their ice velocity products. Financial support is provided by the National



Key R & D Program of China (2017YFA0603103), the National Natural Science
Foundation of China (Grant Nos. 41431070, 41590854 and 41621091), the Key
Research Program of Frontier Sciences, CAS (Grant Nos. QYZDB-SSW-DQC027 and
QYZDJ-SSW-DQC042), NASA (Grant No. NNX10AG31G), and the "Strategic
Priority Research Program" of the Chinese Academy of Sciences(XDA19070302). We
also appreciate the efforts of Amelie Driemel, the Data Publisher for Earth &
Environmental Science, toward archiving the data at the World Data Center
PANGAEA.

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
