# Peer review of "Present-day high-resolution ice velocity map of the Antarctic ice sheet"

_Earth System Science Data, 2018_

## Referee Comment (RC1) · Anonymous Referee #1 · 9 Apr 2019

Shen et al. present a nearly complete ice velocity map of the Antarctic Ice sheet obtained by mosaicking individual velocity measurements over a period of 3 summers. The velocity measurements are validated against sparse in situ measurements and also compared to a similar ice velocity map derived using the INSAR technique.

The degree of novelty of this article is low. The same group of authors already published a paper last year using and presenting this velocity data, the validation is not really convincing because they compare to in situ measurements made several decades ago and they totally ignored in their comparison a published Antarctic-wide velocity map derived using the same images (Landsat8).

General comments

1/ The same authors already published a paper using these data in Scientific Reports (https://www.nature.com/articles/s41598-018-22765-0) so the degree of novelty is, at best, incremental. I have much more expectation for an ESSD data in term of originality. More worrysome is the fact that some sections of the submitted manuscript are "copy and paste" from the supplementary text from this earlier publication. I suggest that the editor asks the EGU Copernicus office to produce, if possible, a similarity index between the present article and the supplement of that published study to back up and quantify my comment. The fact that the authors did not cite their earlier work (was it deliberate?) also raises an ethical problem.

2/ I find it problematic to have a dataset provided in the format of a commercial software (ENVI). Not the SPIRIT of ESSD I feel. Geotiff raster file should be prefered as they can be read by mean different software open source and commercial.

3/ Error assessment. Authors start with a formal error description identifying all sources of errors and then change strategy and just assess the error as the spread in their different velocity fields. But all their velocity fields have priorly been adjusted to the same reference, the slow moving ice in the INSAR velocity mosaic from Mouginot/Rignot, so the spread is, by construction, reduced. If the INSAR mosaic is wrong in these slow moving areas (it includes some artefacts), then all L8 velocity fields will also be wrong. The spread between them will remain small but it does not mean they are accurate.

4/ Velocity maps based on Landsat 8 imagery have also been published by another group of authors (Gardner et al., TC, 2018) and a thorough comparison to their results is mandatory. https://www.the-cryosphere.net/12/521/2018/

Specific comments

Title : "present day" is too general. Provide dates Time stamp. Problematic to average over 3 summers. Is not it possible to have 3 maps (one per summer) instead (or as an additional product). Would be more useful to user even if they are not complete. Robust internal and external validation is too vague. Be more specific L25 "ice glacier"

[Figure]

is not an appropriate terminology Many space missing before the references in the text L41. The 1986 reference is outdated and probably not the best on the topic. L54. It is not "difficult". It is simply impossible... L56. Missing references about velocity mapping in Antarctica : Bindschadler & Scambos, Science, 1991 ; Scambos et al. RSE 1992 [I see one of them is cited later] L61-67. The flow of the introduction is not OK. At these lines authors come back to the reasons why the velocity data are needed. Improve the structure/logics. L70-71. Cite only reference in Antarctica here. L76. "most recent" not that "new" now... L80. Heid and Kaab did not really work on Antarctica. Cite rather Gardner et al. TC 2018 here L109-111. Already stated. Avoid such repetition and go to the point. L120. "Electronic distance" is a strange method for velocity measurements. L133. Images with time span of 16 days to 3 years are used. What about the seasonal velocity variations? Can they be neglected? L139. Coregistration and correlation are not synonym. They are are two successive steps to generate accurate displacements. L152. This is a complete mis-understanding of what cosi corr is doing. The coregistration is the step to obtain two images without any shift on the stable terrain (if stable terrain exist in the images). This is not the result of the first, coarse correlation. It gives the impression that the authors did not understand the tool that they have been using. Worrysome. L159. 32 pixels means 480 m for the smaller correlation window. With such a window, does it make sense to generate a final map with a sampling distance of 100m? Authors should quantify what is the actual/true resolution of their dataset and use a relevant final grid size. Ground sampling distance and horizontal resolution are two different concepts. L179. Unclear how the QA band has been generated. The info needs to be provided. L188. This is too vague. Which value are exactly excluded? L201. This is this geolocation error that the coregistration aim at reducing. I understand it is difficult (or impossible) in Antarctica due to a lack of stable terrain (few nunataks). But I write this to make sure the authors understand what the coregistration step is. L211. How can the authors be sure that the INSAR dataset provide a good reference for slow moving areas? I noted many artefacts in the low velocity zone of Antarctica in the INSAR dataset. L232.

Does it mean that more weight is given to the velocity measurements over longer time periods? Authors should be more explicit about that. L247. Your ground sampling distance is finer. Authors did not demonstrate that the actual resolution of their product if finer. To be demonstrated. Authors need to show side by side their dataset with the Rignot & Gardner dataset to really illustrate their high resolution. L255. Clarify that "fast ice" is on sea ice. Not obvious to all readers. L297. What does "predominant" mean? No really scientific Table 1. Why providing the "year" if it is always 2015... L315. Why is the validation called "technical"? L317-321 example of sentences with redundant (or incoherent information). L332. Here the initial definition of co-registration is correct. Not earlier in the text. However, decorrelation is not leading to bad coregistration. Again the authors seem to not understand the meaning of this word. L342. Authors are quoting a correlation error here, not a coregistration error L350. This is true only if the velocity is not changing with time. How can the authors be sure? L362. Constance of the coregistration error with space and time need to be demonstrated or backed up using a formula. L383. This variability of glacier flow is excatly why their mosaicking strategy (three summer) is not appropriate. L385. Is there a year for the Raup & Scambos référence? L399. What about the Gardner et al. product? How does it compare to the same data points? L420. Why not a direct 1:1 comparison between the INSAR and L8 dataset (also in area flowing between 10 and 100 m/yr)? L426. Validation on fast flowing glaciers, using data acquired 30 years ago in areas which are known to potentially experience profound change in velocity is not appropriate. I think it would be more convincing to construct two or three annual velocity maps and compare them. Over one year the assumption of little velocity change has more chance to hold. It would a useful check of internal consistency.
* * *

---

## Author Comment (AC1) · 1 May 2019

**1 Dear Referee:**

- 2 Thanks for your careful prompt response on evaluating our manuscript. We appreciate your
- 3 suggestions and valuable comments, which, we believe, have improved our manuscript. We
- 4 have carefully considered your comments and revised the manuscript, including the addition
- 5 of LISA ice speed data for comparison, redrawing Figures 3-7 and the addition of Figure 8, and
- 6 new context for internal consistency of annual ice velocity data, etc. In the following, we
- 7 provide point-by-point response to your comments and concerns. For clarity, our responses
- 8 to the comments are marked in blue.
- 9 Sincerely,
- 10 Dr. Qiang Shen
- 11 Institute of Geodesy & Geophysics, CAS
- 12 Wuhan, China
- 13 cl980606@whigg.ac.cn
- 14

**15 Reply to the Referee #1**

**16 Point-by-Point Response**

Q. ...The degree of novelty of this article is low. The same group of authors already
 published a paper last year using and presenting this velocity data, the validation is not
 really convincing because they compare to in situ measurements made several decades ago
 and they totally ignored in their comparison a published Antarctic-wide velocity map
 derived using the same images (Landsat8).

R. Thank you for your comment. We respectfully disagree on your comment on the low 23 24 novelty of our manuscript. Our published Nature Scientific Report Paper on March 2018 using 25 a previous version of Landsat 8 based Antarctica-wide ice velocity data, which were used to 26 articulate the findings of ice mass loss in Antarctica. In that publication, the details of the 27 prior version of the ice velocity products was concisely described in the supplemental 28 material. Here and in this ESSD manuscript, we describe an updated, validated, and improved 29 version of Antarctica-wide ice velocity data products, including more paired displacement 30 vectors, and improved post-processing method, etc. We believe our manuscript fits ESSD 31 journal's scientific criteria to provide a new, comprehensive data product to the glaciological 32 community for anticipated additional scientific studies.

On the validation of our data product, I am afraid we have made exhaustive studies. In 35 Antarctica, verification of satellite-derived measurements is a challenging task because of its 36 remoteness and extremely cold climate. It is extremely difficult to collect field data 37 synchronously in Antarctica. In the manuscript, we collected nearly all field data for the 38 validation, which is obviously better than those of the previously published papers. Although 39 these in-situ data were measured decadal years ago, they are reasonable for the validation 40 purpose since the ice velocity is relatively stable, especially for the interior of the ice sheet. 41 In addition, for this time we compare our ice velocity results with the LISA750 data according 42 to the Reviewer's advice, the details can be found in main text and following the responses.

- 43
- 44 2. Q. General comments

1/ The same authors already published a paper using these data in Scientific Reports 46 (https://www.nature.com/articles/s41598-018-22765-0) so the degree of novelty is, at best, 47 incremental. I have much more expectation for an ESSD data in term of originality. More 48 worrysome is the fact that some sections of the submitted manuscript are "copy and paste" 49 from the supplementary text from this earlier publication. I suggest that the editor asks the 50 EGU Copernicus office to produce, if possible, a similarity index between the present article 51 and the supplement of that published study to back up and quantify my comment. The fact 52 that the authors did not cite their earlier work (was it deliberate?) also raises an ethical 53 problem.

R. As stated in response 1, the work is motivated by the purpose of providing a new high-56 quality data to reuse for glaciological community which is very relevant to the topic of the 57 ESSD journal. The similarity report had been completed before the stage of the open 58 discussion, which our previous paper has been included . We respectfully disagree with the 59 Reviewer that our paper lacks novelty. We undertook an improved post-processing method 60 based on more paired displacement vectors, in the generation of the new and improved 61 Antarctica-wide ice velocity data product. In addition, we had cited our earlier work in dataset 62 repository (htttps://doi.pangaea.de/10.1594/PANGAEA.895738). The earlier work was not 63 cited in main text because we were preparing these two manuscripts at the same time and 64 the earlier work focused on the scientific findings in Antarctic ice sheet. Thanks for your 65 suggestions, we now have cited our earlier work in the abstract and introduction and results sections of the main text. The details can be found in revised manuscript. 66

3.Q. 2/I find it problematic to have a dataset provided in the format of a commercial software
(ENVI). Not the SPIRIT of ESSD I feel. Geotiff raster file should be prefered as they can be read
by mean different software open source and commercial.

R. Thank for your comments. In fact, the datasets are provided in the binary format including
a header file, which can be opened not just by the commercial software. Anyway, we have
updated the datasets in Geotiff format for users to easily read or open the data files.

4. Q. 3/Error assessment. Authors start with a formal error description identifying all sources
of errors and then change strategy and just assess the error as the spread in their different
velocity fields. But all their velocity fields have priorly been adjusted to the same reference,
the slow moving ice in the INSAR velocity mosaic from Mouginot/Rignot, so the spread is, by
construction, reduced. If the INSAR mosaic is wrong in these slow moving areas (it includes
some artefacts), then all L8 velocity fields will also be wrong. The spread between them will
remain small but it does not mean they are accurate.

R. Precise assessment of ice velocity datasets is crucial. However, it is challenging task in 85 Antarctica because its remoteness and harsh climate and environment. The gridded error map 86 of ice velocity is based on a given precision of image coregistration as did in the same manner 87 by Mouginot (2017) /Rignot (2011) and Gardner (2018) For accuracy assessment, we also 88 compared our results with the released ice velocity datasets derived from satellite images and 89 field surveying data. However, there are different time epochs, precise assessment is still 90 difficult. We first investigated the consistence between our results and InSAR, and LISA ice 91 velocity data is now also used for the similar work. The eight profiles of ice velocity datasets 92 show very good consistence (see Figure 3). Second, we investigated our performance of our 93 results by comparing to field surveying measurements. Although these measurements were 94 collected several decades ago, such comparisons are still thought to be reasonable since the ice velocity is slow and stable especially for the interior of the ice sheet, which has been96 supported by the final comparison results (Figures 5 and 6).

For absolute calibration of ice velocity. Here, we first used the rock data as reference sources, 99 however, it is still difficult to find enough rock places in the interior of ice sheet, therefore, we 100 also used existing SAR ice velocity as a reference to define stable areas (ice velocity
|------------------------------------------------------------------------------------------------------------------------------------------------------------------------------------|-----------------------------------------------------------------------------------------------------------------------------------------------------------------------------------------------------------------------------------------------------------------------------------------------------------------------------------------------------------------------------------------------------------------------------------------------------------------------------------------------------------------------------------------------------------------------------------------------------------------------------------------------------------------------------------------------------------------------------------------------------------------------------------------------------------------------------------------------------------------------------------------------------------------------------------------------------------------------------------------------------------------------------------------------------------------------------------------------------------------------------------------------------------------------------------------------|
| 150                                                                                                                                                                                | R: Thank you for your comment. They have been changed, and new references have been                                                                                                                                                                                                                                                                                                                                                                                                                                                                                                                                                                                                                                                                                                                                                                                                                                                                                                                                                                                                                                                                                                           |
| 150                                                                                                                                                                                | replaced by APE chapter 4 and Jan Joughin et al. 2011                                                                                                                                                                                                                                                                                                                                                                                                                                                                                                                                                                                                                                                                                                                                                                                                                                                                                                                                                                                                                                                                                                                                         |
| 151                                                                                                                                                                                | replaced by ARS chapter 4 and lan joughin et al., 2011.                                                                                                                                                                                                                                                                                                                                                                                                                                                                                                                                                                                                                                                                                                                                                                                                                                                                                                                                                                                                                                                                                                                                       |
| 152                                                                                                                                                                                |                                                                                                                                                                                                                                                                                                                                                                                                                                                                                                                                                                                                                                                                                                                                                                                                                                                                                                                                                                                                                                                                                                                                                                                               |
| 153                                                                                                                                                                                | <ol> <li>Q L54. It is not "difficult". It is simply impossible</li> </ol>                                                                                                                                                                                                                                                                                                                                                                                                                                                                                                                                                                                                                                                                                                                                                                                                                                                                                                                                                                                                                                                                                                                     |
| 154                                                                                                                                                                                |                                                                                                                                                                                                                                                                                                                                                                                                                                                                                                                                                                                                                                                                                                                                                                                                                                                                                                                                                                                                                                                                                                                                                                                               |
| 155                                                                                                                                                                                | R: Changed according to your advice, thanks.                                                                                                                                                                                                                                                                                                                                                                                                                                                                                                                                                                                                                                                                                                                                                                                                                                                                                                                                                                                                                                                                                                                                                  |
| 156                                                                                                                                                                                |                                                                                                                                                                                                                                                                                                                                                                                                                                                                                                                                                                                                                                                                                                                                                                                                                                                                                                                                                                                                                                                                                                                                                                                               |
| 157                                                                                                                                                                                | 12 O: 156 Missing references about velocity manning in Antarctica - Rindschadler & Scambos                                                                                                                                                                                                                                                                                                                                                                                                                                                                                                                                                                                                                                                                                                                                                                                                                                                                                                                                                                                                                                                                                                    |
| 157                                                                                                                                                                                | 12. Q. ESO. Missing references about velocity mapping in Antal citica : binuschauler & Scambos,                                                                                                                                                                                                                                                                                                                                                                                                                                                                                                                                                                                                                                                                                                                                                                                                                                                                                                                                                                                                                                                                                               |
| 158                                                                                                                                                                                | Science, 1991; Scambos et al. RSE 1992 [I see one of them is cited later]                                                                                                                                                                                                                                                                                                                                                                                                                                                                                                                                                                                                                                                                                                                                                                                                                                                                                                                                                                                                                                                                                                                     |
| 159                                                                                                                                                                                |                                                                                                                                                                                                                                                                                                                                                                                                                                                                                                                                                                                                                                                                                                                                                                                                                                                                                                                                                                                                                                                                                                                                                                                               |
| 160                                                                                                                                                                                | R: The missing references have been now included, they are Bindschadler & Scambos, Science,                                                                                                                                                                                                                                                                                                                                                                                                                                                                                                                                                                                                                                                                                                                                                                                                                                                                                                                                                                                                                                                                                                   |
| 161                                                                                                                                                                                | 1991; Scambos et al. RSE 1992, Scheuchl 2012. Thanks.                                                                                                                                                                                                                                                                                                                                                                                                                                                                                                                                                                                                                                                                                                                                                                                                                                                                                                                                                                                                                                                                                                                                         |
| 162                                                                                                                                                                                |                                                                                                                                                                                                                                                                                                                                                                                                                                                                                                                                                                                                                                                                                                                                                                                                                                                                                                                                                                                                                                                                                                                                                                                               |
| 163                                                                                                                                                                                |                                                                                                                                                                                                                                                                                                                                                                                                                                                                                                                                                                                                                                                                                                                                                                                                                                                                                                                                                                                                                                                                                                                                                                                               |
| 161                                                                                                                                                                                | 12 OIL61 67 The flow of the introduction is not OK. At these lines outhers some back to the                                                                                                                                                                                                                                                                                                                                                                                                                                                                                                                                                                                                                                                                                                                                                                                                                                                                                                                                                                                                                                                                                                   |
| 104                                                                                                                                                                                | 15. Q. LOI-07. The now of the introduction is not OK. At these lines authors come back to the                                                                                                                                                                                                                                                                                                                                                                                                                                                                                                                                                                                                                                                                                                                                                                                                                                                                                                                                                                                                                                                                                                 |
| 165                                                                                                                                                                                | reasons why the velocity data are needed. Improve the structure/logics.                                                                                                                                                                                                                                                                                                                                                                                                                                                                                                                                                                                                                                                                                                                                                                                                                                                                                                                                                                                                                                                                                                                       |
| 166                                                                                                                                                                                |                                                                                                                                                                                                                                                                                                                                                                                                                                                                                                                                                                                                                                                                                                                                                                                                                                                                                                                                                                                                                                                                                                                                                                                               |
| 167                                                                                                                                                                                | R: Sorry for misleading. Thank you for your advice. we edit the part and move it into the back                                                                                                                                                                                                                                                                                                                                                                                                                                                                                                                                                                                                                                                                                                                                                                                                                                                                                                                                                                                                                                                                                                |
| 168                                                                                                                                                                                | of the first paragraph in introduction . See L47-50 in revised manuscript.                                                                                                                                                                                                                                                                                                                                                                                                                                                                                                                                                                                                                                                                                                                                                                                                                                                                                                                                                                                                                                                                                                                    |
| 169                                                                                                                                                                                |                                                                                                                                                                                                                                                                                                                                                                                                                                                                                                                                                                                                                                                                                                                                                                                                                                                                                                                                                                                                                                                                                                                                                                                               |
| 170                                                                                                                                                                                | 14 O. 170-71 Cite only reference in Antarctica here                                                                                                                                                                                                                                                                                                                                                                                                                                                                                                                                                                                                                                                                                                                                                                                                                                                                                                                                                                                                                                                                                                                                           |
| 170                                                                                                                                                                                | 14. Q: 170-71. Cite only reference in Antarctica here.                                                                                                                                                                                                                                                                                                                                                                                                                                                                                                                                                                                                                                                                                                                                                                                                                                                                                                                                                                                                                                                                                                                                        |
| 1/1                                                                                                                                                                                | R: OK, It has been changed, the References, Joughin et al. 2002, Scambos et al., 1992, Scheuchi                                                                                                                                                                                                                                                                                                                                                                                                                                                                                                                                                                                                                                                                                                                                                                                                                                                                                                                                                                                                                                                                                               |
| 172                                                                                                                                                                                | et al.,2012 have been added.                                                                                                                                                                                                                                                                                                                                                                                                                                                                                                                                                                                                                                                                                                                                                                                                                                                                                                                                                                                                                                                                                                                                                                  |
| 470                                                                                                                                                                                |                                                                                                                                                                                                                                                                                                                                                                                                                                                                                                                                                                                                                                                                                                                                                                                                                                                                                                                                                                                                                                                                                                                                                                                               |
| 1/3                                                                                                                                                                                |                                                                                                                                                                                                                                                                                                                                                                                                                                                                                                                                                                                                                                                                                                                                                                                                                                                                                                                                                                                                                                                                                                                                                                                               |
| 173
                                                                                                                                                               | 15. Q: L76. "most recent" not that "new" now                                                                                                                                                                                                                                                                                                                                                                                                                                                                                                                                                                                                                                                                                                                                                                                                                                                                                                                                                                                                                                                                                                                                                  |
| 173
                                                                                                                                                        | 15. Q: L76. "most recent" not that "new" now                                                                                                                                                                                                                                                                                                                                                                                                                                                                                                                                                                                                                                                                                                                                                                                                                                                                                                                                                                                                                                                                                                                                                  |
| 173
                                                                                                                                                 | 15. Q: L76. "most recent" not that "new" now
R: Thanks, it has been changed.                                                                                                                                                                                                                                                                                                                                                                                                                                                                                                                                                                                                                                                                                                                                                                                                                                                                                                                                                                                                                                                                                                               |
| 173
                                                                                                                                          | 15. Q: L76. "most recent" not that "new" now
R: Thanks, it has been changed.                                                                                                                                                                                                                                                                                                                                                                                                                                                                                                                                                                                                                                                                                                                                                                                                                                                                                                                                                                                                                                                                                                               |
| 173
                                                                                                                                   |  <li>15. Q: L76. "most recent" not that "new" now</li> <li>R: Thanks, it has been changed.</li> <li>16. Q: L80. Heid and Kaab did not really work on Antarctica. Cite rather Gardner et al. TC 2018.</li>                                                                                                                                                                                                                                                                                                                                                                                                                                                                                                                                                                                                                                                                                                                                                                                                                                                                                                                                                                            |
| 173
                                                                                                                                   |  <li>15. Q: L76. "most recent" not that "new" now</li> <li>R: Thanks, it has been changed.</li> <li>16. Q:L80. Heid and Kaab did not really work on Antarctica. Cite rather Gardner et al. TC 2018 here.</li>                                                                                                                                                                                                                                                                                                                                                                                                                                                                                                                                                                                                                                                                                                                                                                                                                                                                                                                                                                        |
| 173
                                                                                                                            | <li>15. Q: L76. "most recent" not that "new" now</li><li>R: Thanks, it has been changed.</li><li>16. Q:L80. Heid and Kaab did not really work on Antarctica. Cite rather Gardner et al. TC 2018 here</li>                                                                                                                                                                                                                                                                                                                                                                                                                                                                                                                                                                                                                                                                                                                                                                                                                                                                                                                                                                            |
| 173
                                                                                                                     |  <li>15. Q: L76. "most recent" not that "new" now</li> <li>R: Thanks, it has been changed.</li> <li>16. Q:L80. Heid and Kaab did not really work on Antarctica. Cite rather Gardner et al. TC 2018 here</li>                                                                                                                                                                                                                                                                                                                                                                                                                                                                                                                                                                                                                                                                                                                                                                                                                                                                                                                                                                         |
| 173
                                                                                                              |  <li>15. Q: L76. "most recent" not that "new" now</li> <li>R: Thanks, it has been changed.</li> <li>16. Q:L80. Heid and Kaab did not really work on Antarctica. Cite rather Gardner et al. TC 2018 here</li> <li>R: Thanks, it has been changed.</li>                                                                                                                                                                                                                                                                                                                                                                                                                                                                                                                                                                                                                                                                                                                                                                                                                                                                                                                                |
| 173
                                                                                                       |  <li>15. Q: L76. "most recent" not that "new" now</li> <li>R: Thanks, it has been changed.</li> <li>16. Q:L80. Heid and Kaab did not really work on Antarctica. Cite rather Gardner et al. TC 2018 here</li> <li>R: Thanks, it has been changed.</li>                                                                                                                                                                                                                                                                                                                                                                                                                                                                                                                                                                                                                                                                                                                                                                                                                                                                                                                                |
| 173
                                                                                                |  <li>15. Q: L76. "most recent" not that "new" now</li> <li>R: Thanks, it has been changed.</li> <li>16. Q:L80. Heid and Kaab did not really work on Antarctica. Cite rather Gardner et al. TC 2018 here</li> <li>R: Thanks, it has been changed.</li> <li>17. Q: L109-111. Already stated. Avoid such repetition and go to the point.</li>                                                                                                                                                                                                                                                                                                                                                                                                                                                                                                                                                                                                                                                                                                                                                                                                                                           |
| 173
                                                                                         |  <li>15. Q: L76. "most recent" not that "new" now</li> <li>R: Thanks, it has been changed.</li> <li>16. Q:L80. Heid and Kaab did not really work on Antarctica. Cite rather Gardner et al. TC 2018 here</li> <li>R: Thanks, it has been changed.</li> <li>17. Q: L109-111. Already stated. Avoid such repetition and go to the point.</li>                                                                                                                                                                                                                                                                                                                                                                                                                                                                                                                                                                                                                                                                                                                                                                                                                                           |
| 173
                                                                                  |  <li>15. Q: L76. "most recent" not that "new" now</li> <li>R: Thanks, it has been changed.</li> <li>16. Q:L80. Heid and Kaab did not really work on Antarctica. Cite rather Gardner et al. TC 2018 here</li> <li>R: Thanks, it has been changed.</li> <li>17. Q: L109-111. Already stated. Avoid such repetition and go to the point.</li> <li>B: Thanks, 'using the optical offset method, which will be summarized in Section 2.3' is deleted.</li>                                                                                                                                                                                                                                                                                                                                                                                                                                                                                                                                                                                                                                                                                                                                |
| 173
                                                                           |  <li>15. Q: L76. "most recent" not that "new" now</li> <li>R: Thanks, it has been changed.</li> <li>16. Q:L80. Heid and Kaab did not really work on Antarctica. Cite rather Gardner et al. TC 2018 here</li> <li>R: Thanks, it has been changed.</li> <li>17. Q: L109-111. Already stated. Avoid such repetition and go to the point.</li> <li>R: Thanks. 'using the optical offset method, which will be summarized in Section 2.3' is deleted.</li>                                                                                                                                                                                                                                                                                                                                                                                                                                                                                                                                                                                                                                                                                                                                |
| 173
                                                                           |  <li>15. Q: L76. "most recent" not that "new" now</li> <li>R: Thanks, it has been changed.</li> <li>16. Q:L80. Heid and Kaab did not really work on Antarctica. Cite rather Gardner et al. TC 2018 here</li> <li>R: Thanks, it has been changed.</li> <li>17. Q: L109-111. Already stated. Avoid such repetition and go to the point.</li> <li>R: Thanks. 'using the optical offset method, which will be summarized in Section 2.3' is deleted.</li>                                                                                                                                                                                                                                                                                                                                                                                                                                                                                                                                                                                                                                                                                                                                |
| 173
                                                                    |  <li>15. Q: L76. "most recent" not that "new" now</li> <li>R: Thanks, it has been changed.</li> <li>16. Q: L80. Heid and Kaab did not really work on Antarctica. Cite rather Gardner et al. TC 2018 here</li> <li>R: Thanks, it has been changed.</li> <li>17. Q: L109-111. Already stated. Avoid such repetition and go to the point.</li> <li>R: Thanks. 'using the optical offset method, which will be summarized in Section 2.3' is deleted.</li> <li>18. Q: L120. "Electronic distance" is a strange method for velocity measurements.</li>                                                                                                                                                                                                                                                                                                                                                                                                                                                                                                                                                                                                                                    |
| 173
                                                             |  <li>15. Q: L76. "most recent" not that "new" now</li> <li>R: Thanks, it has been changed.</li> <li>16. Q:L80. Heid and Kaab did not really work on Antarctica. Cite rather Gardner et al. TC 2018 here</li> <li>R: Thanks, it has been changed.</li> <li>17. Q: L109-111. Already stated. Avoid such repetition and go to the point.</li> <li>R: Thanks. 'using the optical offset method, which will be summarized in Section 2.3' is deleted.</li> <li>18. Q: L120. "Electronic distance" is a strange method for velocity measurements.</li>                                                                                                                                                                                                                                                                                                                                                                                                                                                                                                                                                                                                                                     |
| 173
                                                      |  <li>15. Q: L76. "most recent" not that "new" now</li> <li>R: Thanks, it has been changed.</li> <li>16. Q:L80. Heid and Kaab did not really work on Antarctica. Cite rather Gardner et al. TC 2018 here</li> <li>R: Thanks, it has been changed.</li> <li>17. Q: L109-111. Already stated. Avoid such repetition and go to the point.</li> <li>R: Thanks. 'using the optical offset method, which will be summarized in Section 2.3' is deleted.</li> <li>18. Q: L120. "Electronic distance" is a strange method for velocity measurements.</li> <li>R: it appeared in the 'data acquisition method' of velmap project. Here, we also used the</li>                                                                                                                                                                                                                                                                                                                                                                                                                                                                                                                                  |
| 173
                                               |  <li>15. Q: L76. "most recent" not that "new" now</li> <li>R: Thanks, it has been changed.</li> <li>16. Q:L80. Heid and Kaab did not really work on Antarctica. Cite rather Gardner et al. TC 2018 here</li> <li>R: Thanks, it has been changed.</li> <li>17. Q: L109-111. Already stated. Avoid such repetition and go to the point.</li> <li>R: Thanks. 'using the optical offset method, which will be summarized in Section 2.3' is deleted.</li> <li>18. Q: L120. "Electronic distance" is a strange method for velocity measurements.</li> <li>R: it appeared in the 'data acquisition method' of velmap project. Here, we also used the terminology. The electronic distance is a surveying and mapping technique.</li>                                                                                                                                                                                                                                                                                                                                                                                                                                                       |
| 173
                                        |  <li>15. Q: L76. "most recent" not that "new" now</li> <li>R: Thanks, it has been changed.</li> <li>16. Q:L80. Heid and Kaab did not really work on Antarctica. Cite rather Gardner et al. TC 2018 here</li> <li>R: Thanks, it has been changed.</li> <li>17. Q: L109-111. Already stated. Avoid such repetition and go to the point.</li> <li>R: Thanks. 'using the optical offset method, which will be summarized in Section 2.3' is deleted.</li> <li>18. Q: L120. "Electronic distance" is a strange method for velocity measurements.</li> <li>R: it appeared in the 'data acquisition method' of velmap project. Here, we also used the terminology. The electronic distance is a surveying and mapping technique.</li>                                                                                                                                                                                                                                                                                                                                                                                                                                                       |
| 173
                                 |  <li>15. Q: L76. "most recent" not that "new" now</li> <li>R: Thanks, it has been changed.</li> <li>16. Q:L80. Heid and Kaab did not really work on Antarctica. Cite rather Gardner et al. TC 2018 here</li> <li>R: Thanks, it has been changed.</li> <li>17. Q: L109-111. Already stated. Avoid such repetition and go to the point.</li> <li>R: Thanks. 'using the optical offset method, which will be summarized in Section 2.3' is deleted.</li> <li>18. Q: L120. "Electronic distance" is a strange method for velocity measurements.</li> <li>R: it appeared in the 'data acquisition method' of velmap project. Here, we also used the terminology. The electronic distance is a surveying and mapping technique.</li>                                                                                                                                                                                                                                                                                                                                                                                                                                                       |
| 173
                          |  <li>15. Q: L76. "most recent" not that "new" now</li> <li>R: Thanks, it has been changed.</li> <li>16. Q:L80. Heid and Kaab did not really work on Antarctica. Cite rather Gardner et al. TC 2018 here</li> <li>R: Thanks, it has been changed.</li> <li>17. Q: L109-111. Already stated. Avoid such repetition and go to the point.</li> <li>R: Thanks. 'using the optical offset method, which will be summarized in Section 2.3' is deleted.</li> <li>18. Q: L120. "Electronic distance" is a strange method for velocity measurements.</li> <li>R: it appeared in the 'data acquisition method' of velmap project. Here, we also used the terminology. The electronic distance is a surveying and mapping technique.</li>                                                                                                                                                                                                                                                                                                                                                                                                                                                       |
| 173
                          |  <li>15. Q: L76. "most recent" not that "new" now</li> <li>R: Thanks, it has been changed.</li> <li>16. Q:L80. Heid and Kaab did not really work on Antarctica. Cite rather Gardner et al. TC 2018 here</li> <li>R: Thanks, it has been changed.</li> <li>17. Q: L109-111. Already stated. Avoid such repetition and go to the point.</li> <li>R: Thanks. 'using the optical offset method, which will be summarized in Section 2.3' is deleted.</li> <li>18. Q: L120. "Electronic distance" is a strange method for velocity measurements.</li> <li>R: it appeared in the 'data acquisition method' of velmap project. Here, we also used the terminology. The electronic distance is a surveying and mapping technique.</li> <li>19. Q: L133. Images with time span of 16 days to 3 years are used. What about the seasonal velocity wasintions? Can they be perfected?</li>                                                                                                                                                                                                                                                                                                       |
| 173
                   | 15. Q: L76. "most recent" not that "new" now R: Thanks, it has been changed. 16. Q:L80. Heid and Kaab did not really work on Antarctica. Cite rather Gardner et al. TC 2018 here R: Thanks, it has been changed. 17. Q: L109-111. Already stated. Avoid such repetition and go to the point. R: Thanks. 'using the optical offset method, which will be summarized in Section 2.3' is deleted. 18. Q: L120. "Electronic distance" is a strange method for velocity measurements. R: it appeared in the 'data acquisition method' of velmap project. Here, we also used the terminology. The electronic distance is a surveying and mapping technique. 19. Q: L133. Images with time span of 16 days to 3 years are used. What about the seasonal velocity variations? Can they be neglected?                                                                                                                                                                                                                                                                                                                                                                                                  |
| 173
            |  <li>15. Q: L76. "most recent" not that "new" now</li> <li>R: Thanks, it has been changed.</li> <li>16. Q:L80. Heid and Kaab did not really work on Antarctica. Cite rather Gardner et al. TC 2018 here</li> <li>R: Thanks, it has been changed.</li> <li>17. Q: L109-111. Already stated. Avoid such repetition and go to the point.</li> <li>R: Thanks. 'using the optical offset method, which will be summarized in Section 2.3' is deleted.</li> <li>18. Q: L120. "Electronic distance" is a strange method for velocity measurements.</li> <li>R: it appeared in the 'data acquisition method' of velmap project. Here, we also used the terminology. The electronic distance is a surveying and mapping technique.</li> <li>19. Q: L133. Images with time span of 16 days to 3 years are used. What about the seasonal velocity variations? Can they be neglected?</li>                                                                                                                                                                                                                                                                                                       |
| 173
     |  <li>15. Q: L76. "most recent" not that "new" now</li> <li>R: Thanks, it has been changed.</li> <li>16. Q:L80. Heid and Kaab did not really work on Antarctica. Cite rather Gardner et al. TC 2018 here</li> <li>R: Thanks, it has been changed.</li> <li>17. Q: L109-111. Already stated. Avoid such repetition and go to the point.</li> <li>R: Thanks. 'using the optical offset method, which will be summarized in Section 2.3' is deleted.</li> <li>18. Q: L120. "Electronic distance" is a strange method for velocity measurements.</li> <li>R: it appeared in the 'data acquisition method' of velmap project. Here, we also used the terminology. The electronic distance is a surveying and mapping technique.</li> <li>19. Q: L133. Images with time span of 16 days to 3 years are used. What about the seasonal velocity variations? Can they be neglected?</li> <li>R: Here, we use the time span to obtain the final mosaics of ice velocity. In mosaicking, the</li>                                                                                                                                                                                                |
| 173
|  <li>15. Q: L76. "most recent" not that "new" now</li> <li>R: Thanks, it has been changed.</li> <li>16. Q:L80. Heid and Kaab did not really work on Antarctica. Cite rather Gardner et al. TC 2018 here</li> <li>R: Thanks, it has been changed.</li> <li>17. Q: L109-111. Already stated. Avoid such repetition and go to the point.</li> <li>R: Thanks. 'using the optical offset method, which will be summarized in Section 2.3' is deleted.</li> <li>18. Q: L120. "Electronic distance" is a strange method for velocity measurements.</li> <li>R: it appeared in the 'data acquisition method' of velmap project. Here, we also used the terminology. The electronic distance is a surveying and mapping technique.</li> <li>19. Q: L133. Images with time span of 16 days to 3 years are used. What about the seasonal velocity variations? Can they be neglected?</li> <li>R: Here, we use the time span to obtain the final mosaics of ice velocity. In mosaicking, the seasonal changes are not considered, the results can be considered to the average values over</li>                                                                                                  |
| 173
|  <li>15. Q: L76. "most recent" not that "new" now</li> <li>R: Thanks, it has been changed.</li> <li>16. Q:L80. Heid and Kaab did not really work on Antarctica. Cite rather Gardner et al. TC 2018 here</li> <li>R: Thanks, it has been changed.</li> <li>17. Q: L109-111. Already stated. Avoid such repetition and go to the point.</li> <li>R: Thanks. 'using the optical offset method, which will be summarized in Section 2.3' is deleted.</li> <li>18. Q: L120. "Electronic distance" is a strange method for velocity measurements.</li> <li>R: it appeared in the 'data acquisition method' of velmap project. Here, we also used the terminology. The electronic distance is a surveying and mapping technique.</li> <li>19. Q: L133. Images with time span of 16 days to 3 years are used. What about the seasonal velocity variations? Can they be neglected?</li> <li>R: Here, we use the time span to obtain the final mosaics of ice velocity. In mosaicking, the seasonal changes are not considered, the results can be considered to the average values over three years. But according to your advices, we now also provide the individual season products.</li>  |

20. Q: L139. Coregistration and correlation are not synonym. They are two successive steps to
generate accurate displacements.

R:Yes, the cross-correlation is one of methods to tackle with image registration problem. We
used cross-correlation to stand for the registration in same manner as in the paper of Leprince
et al. (2017). According to your advice, we now use 'coregistration' in introduction of method
and the term 'correlation' is used in description of the techniques. 'or cross-correlation' is
removed. thanks.

21. Q: L152. This is a complete mis-understanding of what cosi corr is doing. The coregistration
is the step to obtain two images without any shift on the stable terrain (if stable terrain exist
in the images). This is not the result of the first, coarse correlation. It gives the impression that
the authors did not understand the tool that they have been using. Worrysome.

R: Please see above. The description about the coregistration of COSI-CORR was directly cited
from the cosi-Corr guide in page 27. In my opinion, displacement can be obtained from the
georeferenced images through two-step correlation under the geographic coordinate system.
So, the first correlation for pixelwise displacement is called as coarse correlation and the
second one for subpixel displacement can be viewed as fine correlation. The terms 'coarse or
fine correlation' are from the InSAR coregistration.

22. Q: L159. 32 pixels means 480 m for the smaller correlation window. With such a window,
does it make sense to generate a final map with a sampling distance of 100m? Authors should
quantify what is the actual/true resolution of their dataset and use a relevant final grid size.
Ground sampling distance and horizontal resolution are two different concepts.

R: In COSI-CORR, the sampling distance is controlled by step size not by Window Size. Here,
we used the 7 pixels as step size and produced nearly 100m ground grids. As an illustration,
we can produce higher resolution displacements as shown in Figure R1. The glacier is
apparently less than 300m in width, but we still produce high resolution displacements as
shown in right part of Figure R1.

Figure R1. A sample of small glacier (left: glacier inventory, and ice velocity as background;
Right: The profiles of ice velocity corresponding to the locations marked by red arrow on the
left, thick line - WE velocity, thin line - NS velocity)

23. Q:. L179. Unclear how the QA band has been generated.

R: The downloaded packaged file of Landsat 8 includes the QA band. But information about
cloud cover is coded in the band, we extracted the cloud coverage information using a
procedure we developed. The QA band has been introduced in L179-187 in the original
manuscript.

24. Q: L188. This is too vague. Which value are exactly excluded?

R: The edges of the displacement vector (image) are clipped in the process of mosaicking. For
more clearly, we re-edit the sentence as follows, "the edges of the displacement vectors are
masked in the process of mosaicking".

25. L201. This is this geolocation error that the coregistration aim at reducing. I understand it
is difficult (or impossible) in Antarctica due to a lack of stable terrain (few nunataks). But I
write this to make sure the authors understand what the coregistration step is

R. Agree. We understand the aim of coregistration step. Thanks.

26. Q: L211. How can the authors be sure that the INSAR dataset provide a good reference for
slow moving areas? I noted many artefacts in the low velocity zone of Antarctica in the INSAR
dataset.

R: In preparation of the manuscript, the published ice velocity is only InSAR dataset. We first used rock data and then used InSAR dataset, and only offset of differences between two datasets are applied in mosaicking. We selected InSAR velocity less than 10 m/yr as stagnant, which is generally under the uncertainty of satellite-derived ice velocity. Not only that, but we investigated their performances by comparing with in-situ measurements. furthermore, in revised manuscript, the gardner's ice velocity dataset is also included in assessment of the accuracy, the details can be found in main text.

27. Q: L232. Does it mean that more weight is given to the velocity measurements over longer270 time periods? Authors should be more explicit about that.

R: Yes, the equation 1 can be rewritten in the following for better clarity,

| 272 | $ \Delta t_1 * \frac{\Delta d_1}{\Delta t_1} + \Delta t_2 * \frac{\Delta d_2}{\Delta t_2} + \dots + \Delta t_n * \frac{\Delta d_n}{\Delta t_n} $ |
|-----|--------------------------------------------------------------------------------------------------------------------------------------------------|
| 275 | $\nu = - \wedge t_1 + \wedge t_2 + \dots + \wedge t_n$                                                                                           |

However, for the concise expression the equation still keeps unchanged.

276

28. Q: L247. Your ground sampling distance is finer. Authors did not demonstrate that the
actual resolution of their product if finer. To be demonstrated. Authors need to show side by
side their dataset with the Rignot & Gardner dataset to really illustrate their high resolution.

R: Our high-resolution results are shown in Figure 2 c as an illustration. Here we show the
resolution comparisons of three groups of results (from our L8, InSAR and LISA) in Figure R2.
In general, there are large fluctuation in velocity in high shear zone, both our result and LISA
data can capture the velocity variations along profiles, but InSAR data do not capture any
velocity changes. It can be easily seen that our result shows finer resolution.

---

## Editor Comment (EC1) · Reinhard Drews (Editor) · 23 May 2019

Dear Authors,

Please accept my sincere apologies for the extremely long review time of your Discussion paper. Reviewer 1 submitted his/her report promptly after I had invited him/her, however, I had large difficulties finding more independent reviewers. I take full responsibility for this delay (and hope that you understand that some of this is not entirely in my hands). I decided to go ahead with an Editor Review. This review will include some of my own comments but also refer to the concerns/answers of the existing review.

Reviewer 1 has raised a number of critical concerns and did not suggest to publish the paper in ESSD. Some (but not all) of the criticism can be traced back to the history of this paper. Nevertheless, publication in ESSD requires a rigorous error analysis and a substantial (and not incremental) novelty compared to previously published datasets. The last point is particularly critical, as the relation between the ESSDD manuscript and the publication in Scientific Reports (Shen et al., 2018) was inadequately described in the initial submission. Some of these shortcomings can be excused, as the papers were likely simultaneously submitted to different journals. However, having a number of near identical paragraphs in both submissions (as revealed by the new similarity report) is a red flag for many scientific journals. As it stands now, I will follow Reviewer 1 and don't recommend publication of the paper.

There is no question that the L8 velocity map provided by the authors was a herculean effort and is an important baseline dataset for many applications in Glaciology. I am also happy to see this done by multiple groups, providing an independent cross-check of the applied methods. Unfortunately, it seems that the bulk part of this achievement has already been accredited for by a previous publication (Shen et al., 2018). If the authors believe they can make a clear case that this dataset is a substantial improvement compared to what has been published by Shen et al. 2018, I invite a response to the criticism raised. Below, I provide some comments/suggestions on how this could be done. However, given that the straight-forward comparison between Shen et al. 2018 and the new product from essd-2018-149 has not been done in the first place, I am not overly confident that this will be successful.

Kind regards,

Reinhard Drews

Line numbers refer to original submission unless stated otherwise.

**Similarity between essd-2018-149 and Shen et al., 2018, Nature Scientific Reports**

The initial similarity report of essd-2018-149 (Dec. 06) did not come up with significant overlap compared to other publications. However, I requested a new similarity report and now the overlap with the Shen et al. 2018 paper is significant. Flagged paragraphs are for example starting in l.51, l. 81, l. 136 and elsewhere. I understand that both papers cover the same scientific topic and use similar methods, however, using weakly paraphrased but in essence the same paragraphs in both papers is not the way to deal with this. Reviewer 1 is correct in pointing that out. Methods that are similar between both papers should be

succinctly mentioned and outsourced in the new paper with a reference to the older paper. Differences, on the other hand, should be explicitly stated and indicate the improvement over the previous approaches.

Some statements appearing word for word in both papers (e.g. "These velocity data have the highest spatial resolution of 100m achieved to date..") are mutually exclusive (only one of the two papers can claim the highest spatial resolution).

Again, I understand that these kind of things happen when papers are submitted simultaneously. However, similarity reports which show a high similarity index are a reason for ESSD to reject papers up front. I apologize, that the initial report did not point that out (supposedly because the Shen et al. 2018 paper was just published at the time). If a revision is attempted the significant overlap between both papers must be removed completely.

**Novelty of the essd-2018-149 compared to Shen et al., 2018, Nature Scientific Reports**

As pointed out above, I see value in multiple groups deriving baseline datasets used by an entire discipline. I, therefore, don't expect necessarily that the authors present significant improvements/differences compared to, for example, the Gardner et al. 2018 dataset. A thorough comparison is required and I appreciate the replies that you have provided regarding this point.

However, the replies regarding the differences between essd-2018-149 and the data presented in Shen et al. 2018 do not yet have enough detail. Examples are:

Section 2.3 Largely reiterates (flagged by similarity report) the processing strategy already outline by Shen et al. 2018. However, the general strategy here should only briefly be mentioned, and differences more highlighted.

Section 2.4 same as above (also flagged by similarity report)

The Methods section would need a dedicated and detailed paragraph highlighting the differences between essd-2018-149 and Shen et al. 2018. I understand that more displacement vectors were used, and also that the post-processing has changed. However, (1) what are the processing changes in detail, and (2) what is their impact on the velocity map presented here compared to Shen et al. 2018? Improvements and differences between both datasets must also be detailed in the results. Your proposed revisions (e.g. l. 616 of revised paper) are not enough to answer this point.

**Validation / Error Analysis**

Figure 4 in essd-2018-149 and Figure S2 in Shen et al. 2018 seem to show the same information content (but with different colorscales). Is that correct? What additional information is provided here? Inclusion of the LISA error maps in the revise version is ok, but more importantly differences to Shen et al. 2018 must be shown.

Section 4.2: I appreciate your effort of assembling ground-truth data for an external validation, and yes I understand it is difficult to deal with the different timings. However, the comparison with in-situ GPS data has already (in parts?) been presented by Shen et al. 2018 (Supplementary Information l. 97). What additional data has been assembled for validation in this study? This needs to be explicitly stated, and results from the previous study should not be reiterated here.

**Minor Comments:**

l. 290: I agree with your comment that ENVI binary files are not a proprietary format and can be read by many programs. However, the majority of scientists is probably more used to GTiffs etc. and providing different file formats may increase usage of the data.

l. 344 double reference Leprince et al. 2017

l. 350 stacking reduced random but not systematic errors.

l. 358 explain n in formula (amount of data available for stacking).